



# Mapping soil moisture across the UK: assimilating cosmic-ray neutron sensors, remotely-sensed indices, rainfall radar and catchment water balance data in a Bayesian hierarchical model

Peter Levy[1] and COSMOS-UK team[2]

[1]Centre for Ecology and Hydrology, Bush Estate, Penicuik, Midlothian, EH26 0QB, United Kingdom
[2]Centre for Ecology and Hydrology, Maclean Building, Benson Lane, Crowmarsh Gifford, Wallingford, Oxfordshire OX10 8BB, United Kingdom. A full list of authors appears at the end of the paper.

**Correspondence:** Peter Levy (plevy@ceh.ac.uk)

**Abstract.** Soil moisture is important in many hydrological and ecological processes. However, data sets which are currently available have issues with accuracy and resolution. To translate remotely-sensed data to an absolute measure of soil moisture requires mapped estimates of soil hydrological properties and estimates of vegetation properties, and this introduces considerable uncertainty. We present an alternative methodology for producing daily maps of soil moisture over the UK at 2-km resolution ("SMUK"). The method is based on a simple empirical model, calibrated with five years of daily data from cosmic-ray neutron sensors at ~40 sites across the country. The model is driven by precipitation, humidity, a remotely-sensed "soil water index" satellite product, and soil porosity. The model explains around 70 % of the variance in the daily observations. The spatial variation in the parameter describing the soil water retention (and thereby the response to precipitation) was estimated using daily water balance data from ~1200 catchments with good coverage across the country. The model parameters were estimated by Bayesian calibration using a Markov chain Monte Carlo method, so as to characterise the posterior uncertainty in the parameters and predictions. We found that the simple model could emulate the behaviour of a more complex process-based model. Given the high resolution of the inputs in time and space, the model can predict the very detailed variation in soil moisture which arises from the sporadic nature of precipitation events, including the small-scale and short-term variations associated with orographic and convective rainfall. Predictions over the period 2016 to 2023 demonstrated realistic patterns following the passage of weather fronts and prolonged droughts. The model has negligible computation time, and inputs and predictions are updated daily, lagging approximately one week behind real time.

## 1 Introduction

Soil moisture is an important controlling variable in many hydrological and ecological processes. In modelling flood risk, the soil saturation status is important in predicting runoff and the catchment response to precipitation (Ahlmer et al., 2018; Chifflard et al., 2018). It affects plant growth, crop yield, and irrigation needs. It also determines the aerobic status of the soil, and thereby the balance between different microbial processes such as nitrification and denitrification, and methanogenesis and methanotrophy (and thus the emissions of the greenhouse gases nitrous oxide and methane, (Davidson, 1992; Zou et al.,



2022). The availability of soil moisture data at high resolution in near-real time therefore has many potential applications in environmental science.

25     In terms of ground-based observations of soil moisture, the UK has a network of around 40 sensors based on the cosmic-ray technique (the COSMOS-UK network, Evans et al., 2016; Stanley et al., 2019). The sensors have a large footprint (integrating over an area a few hundred metres in diameter), are well calibrated to provide absolute soil moisture, and are available in near-real time via telemetry at sub-daily resolution. One approach to providing national-scale soil moisture maps would be to extrapolate the COSMOS network data to the wider UK domain with a geostatistical method. However, 40 sites is small in 30   relation to the area of the UK, the sites are widely spaced compared to the fine scale of variability in soil moisture, and there are no sites in the NW quadrant of the UK.

    Several satellites products are available with global or continental-scale coverage. However, they typically suffer from problems of coarse resolution, low frequency, and limited accuracy (Escorihuela and Quintana-Seguí, 2016; Kolassa et al., 2017; Deng et al., 2020; Räsänen et al., 2022). Critically, the microwave emissivity of soils is determined by the dielectric permit-35   tivity, a function of soil texture and structural properties, as well as soil moisture (Dobson et al., 1985). Furthermore, satellite observations are sensitive to changes in the total water content at the land surface, including water in vegetation, are influenced by surface temperature (Wigneron et al., 2003), and are unreliable in areas of organic soils (Peng et al., 2021; Räsänen et al., 2022). The interpretation of remotely-sensed data therefore depends strongly on mapped estimates of other soil and vegetation properties, and this introduces considerable uncertainty (Caubet et al., 2019). A pragmatic approach is to treat the data as 40   a relative index on an unknown scale, rather than absolute soil moisture, often scaled between the long-term maximum and minimum values recorded in each pixel (Bauer-Marschallinger et al., 2018).

    The UK has a dense network of river discharge monitoring stations, which gives daily information extending back many decades at sites all across the country. Combined with precipitation data, this gives information on dynamic changes in soil water storage in over 1000 catchments. However, for several reasons, the change in soil water storage cannot be directly 45   translated into a change in volumetric soil moisture.

    This paper describes a method which aims to incorporate these various sources of information to make the most accurate mapped estimates of soil moisture as possible. The method is simple statistically, but implemented in a Bayesian way to propagate the uncertainty in an appropriate way.

### 1.1 Modelling soil moisture over time

We start by considering the temporal variation in soil moisture at a point. Soil moisture is a storage term in the water balance equation:

$$\Delta S = P - Q - E \tag{1}$$

where $\Delta S$ is the change in soil water storage over some time interval (typically one day), $P$ is the precipitation rate, $Q$ is the rate of drainage and runoff, and $E$ is the evapotranspiration rate. All of these terms are expressed as a volume of water over





a given catchment area per unit time, i.e. a depth of water per day. To express as a volume of water per unit volume of soil, we need to account for the effective depth for soil water storage over the catchment, $z$, such that $\Delta\theta = \Delta S/z$, where $\Delta\theta$ is the change in volumetric soil moisture.

In terms of system dynamics, $P$ provides a series of sporadic inputs (rain events), whilst $E + Q$ provides a much more continuous ouput, roughly proportional to $\theta$. The time series of $\theta$ can be considered as a number of pulse-decay curves, one for

each rain event, but which may overlap each other. Because both $E$ and $Q$ depend on $\theta$, we can approximate the loss of soil moisture following rain events as exponential decay. From a modelling point of view, this means that temporal variation in $\theta$ is strongly driven by $P$, rather than by $E$ or $Q$, but the relationship is not linear, because the dependence is with the history of precipitation in the preceding days (not instantaneous precipitation). We can account for this by applying an exponential moving average (EMA) to the time series of $P$. This gives recent precipitation high weighting, and earlier precipitation gets

exponentially lower weighting. The EMA precipitation at time $t$ is calculated as:

$$\langle P \rangle_t = P_t \times \lambda + \langle P \rangle_{t-1} \times (1 - \lambda) \tag{2}$$

where $\lambda$ is a smoothing coefficient defining the rate of decay, and the angled brackets $\langle\rangle$ denote the EMA filter. Typically, $\lambda = 2/(n+1)$, where $n$ is the number of time points in the moving average window, but a range of $\lambda$ values can be used, to represent soil components with different water retention properties. The EMA filter effectively provides a transformation

to linearise the relationship between precipitation and soil moisture, because the history of precipitation is incorporated in an appropriate way, such that the exponential weighting term accounts for the exponential decay rate. This has previously been shown to provide a robust method for modelling soil moisture (Pan et al., 2003; Albergel and Martin, 2008; Pezij et al., 2019). We can now write a linear model for predicting soil moisture from past precipitation:

$$\theta = \theta_{\max} - \theta'$$

$$\theta' = \beta_0 + \beta_P \langle P \rangle + \epsilon. \tag{3}$$

$\theta_{\max}$, the soil moisture content at saturation, provides a useful reference point because it has a clear physical interpretation - the soil porosity - which can be estimated purely from soil physical properties, as a simple function of particle density and bulk density. Rather than using an arbitrary intercept, we are modelling the deviation in soil moisture from this relatively well-defined maximum value. The simple linear model has a slope, $\beta_P$, which represents the linear dependence of soil moisture on the

exponentially-weighted precipitation, and an intercept $\beta_0$, representing the value of $\theta'$ that would occur with no precipitation over the EMA window. $\epsilon$ represents the residual error term.

Equation (3) provides a first-order approximation, representing exponential decay following precipitation events. Evapotranspiration will also show variability with meteorological conditions - temperature, humidity, windspeed, and net radiation - and terms can be added to the model to account for this as necessary. In addition, satellite-based remotely-sensed soil moisture in-

dices are sensitive to temporal variation due to evaporative conditions, and these can also be added to the model. The strengths





of the remotely-sensed data are its relatively high resolution in space and time, and its link to actual (as opposed to modelled) variations in soil moisture, even if the scaling of the index into absolute units is not known a priori.

## 1.2 Estimating spatial variability in soil hydraulic properties

Soils differ in their hydraulic properties, meaning that two soil columns with the same initial moisture content may lose water
at different rates, and so will have different values of the $\beta_P$ parameter in Eq. (3). Process-based hydrological models attempt to represent the detailed soil physics governing water movement, typically described by parameters which represent the hydraulic conductivity and soil water retention curves (van Genuchten, 1980; Jury and Horton, 2004). These vary with soil texture and porosity, but in a very complex way involving the distribution of pore sizes and phenomena such as horizontal layering and the connectivity of macropores acting as flow pathways (Kosugi, 1996; Wang et al., 2022). For example, a well-structured clay soil,
with aggregates and large pores in between, will behave quite differently to an unstructured clay soil, with few aggregates and small pore spaces, but this difference is not predictable on the basis of soil texture or porosity. To make reliable extrapolations over large domains with physically-based models requires that we know the spatial distribution of the more detailed parameters, but these are poorly known. Furthermore, around a quarter of the UK is covered by organic (peat) and organo-mineral soils (with a substantial peat-like organic layer). These soils cannot be classified by standard soil texture properties (sand/silt/clay)
as peat contains little or no mineral content. Although the van Genuchten (1980) model can sometimes provide an approximate fit for soil water retention curves in peat, this is not always so, and the parameters can be very variable and difficult to predict (Weiss et al., 1998; Hallema et al., 2015; Liu and Lennartz, 2019). Remotely-sensed soil moisture products generally have very limited accuracy in areas with organic soils (Peng et al., 2021; Räsänen et al., 2022).

This uncertainty in the spatial distribution of the hydraulic parameters dominates current attempts to map soil moisture at
large scales. The spatial distribution of soil moisture from current hydrological models (e.g. Robinson et al., 2017; Bell et al., 2018) and satellite products (e.g. Bauer-Marschallinger et al., 2018; Tomer et al., 2016) largely reflects the underlying hydraulic parameters, which come from a small number of sources (e.g. the UK Hydrology of Soil Types (HOST) classification (Boorman et al., 1995), the Harmonised World Soil Database (HWSD, FAO, 2012), SoilsGrid (Hengl et al., 2017)). While these sources may give reasonable estimates of properties such as bulk density and porosity over wide areas, the true spatial distribution of
the more detailed hydraulic parameters (dictated by pore size distribution and macropore connectivity etc.) is very uncertain, and estimates of soil moisture show little coherence.

Here, we take a statistical approach, and empirically relate soil water loss to more readily observable catchment characteristics that correlate with hydraulic properties. We use the extensive data on river discharge in relation to precipitation available from the UK National River Flow Archive (NRFA) to infer the spatial variation in soil hydraulic properties. These data come
from 1212 sites covering all the major catchments in Great Britain. We can therefore extend the linear model to predict soil moisture including this additional information. By comparing the explanatory power of a variety of such models, including a range of meteorological, soil and satellite-derived variables, we can assess what constitutes a parsimonious model.

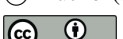



## 1.3 Aims

The work described here aims to develop a method for mapping soil moisture across the UK, based on the highly accurate
COSMOS network data, but incorporating other data sources where these improve predictions. We attempt to answer the
following questions: Can we predict soil moisture with simple linear regression-type models? What variables do we need to
predict soil moisture? Can we incorporate catchment water balance data into large-scale estimates of soil moisture in such a
model? Having derived a parsimonious model, we aim to apply it to predict soil moisture on a 2-km grid across the UK on a
daily basis. Soil moisture predictions carry considerable uncertainty, and we aim to include an appropriate characterisation of
our predictive uncertainty.

## 2 Methods

### 2.1 Data sources

#### 2.1.1 COSMOS network

COSMOS-UK is a network of sites (Fig. 1) equipped with cosmic-ray soil moisture sensors (Evans et al., 2016; Stanley
et al., 2019). The cosmic-ray measurement principle utilizes naturally occurring fast neutrons generated by cosmic rays. These
neutrons interact with water molecules in soil, and the back-scattered flux of slow neutrons is proportional to the soil water
content. The neutron detectors are installed just above the ground, so there is no disturbance to the soil structure. A single
sensor measures a circular footprint with a radius of approximately 100-240 m, and is sensitive to soil moisture in the top
15-30 cm, decreasing exponentially with radial distance (Köhli et al., 2015). A full suite of meteorological measurements are
also made at the sites, including all the variables necessary to calculate potential evapotranspiration. The network provided
61225 observations of daily mean soil moisture, after filtering for data quality and missing values.

#### 2.1.2 Soils data

Soil porosity was estimated using different data sources for Great Britain (GB) and the rest of the domain (Ireland and the edge
of continental Europe). For GB, the UK Countryside Survey has collected bulk density $\rho$ and soil organic carbon measurements
within several hundred 1-km squares since 2007 (Emmett, 2010). Soil organic carbon has been interpolated on to a 1-km grid
covering GB using a generalised additive model (Thomas et al., 2020), and we used this to predict bulk density, using the
relationship between these variables established from the sample data. Porosity was calculated as $1 - \rho/2.65$, where 2.65 is the
standard value for particle density of the solid fraction in g cm$^3$. For the rest of the domain, the same procedure was used, but
using mapped estimates of soil carbon from (de Brogniez et al., 2015). In principle, mapped estimates of bulk density from
(Ballabio et al., 2016) could have been used to estimate porosity more directly, but these did not reflect the range in porosity
seen in organic soils in the UK and Ireland, where values over 70 % are typical. Other variables were obtained from mapped
estimates based on the European LUCAS data (Ballabio et al., 2016, 2019).



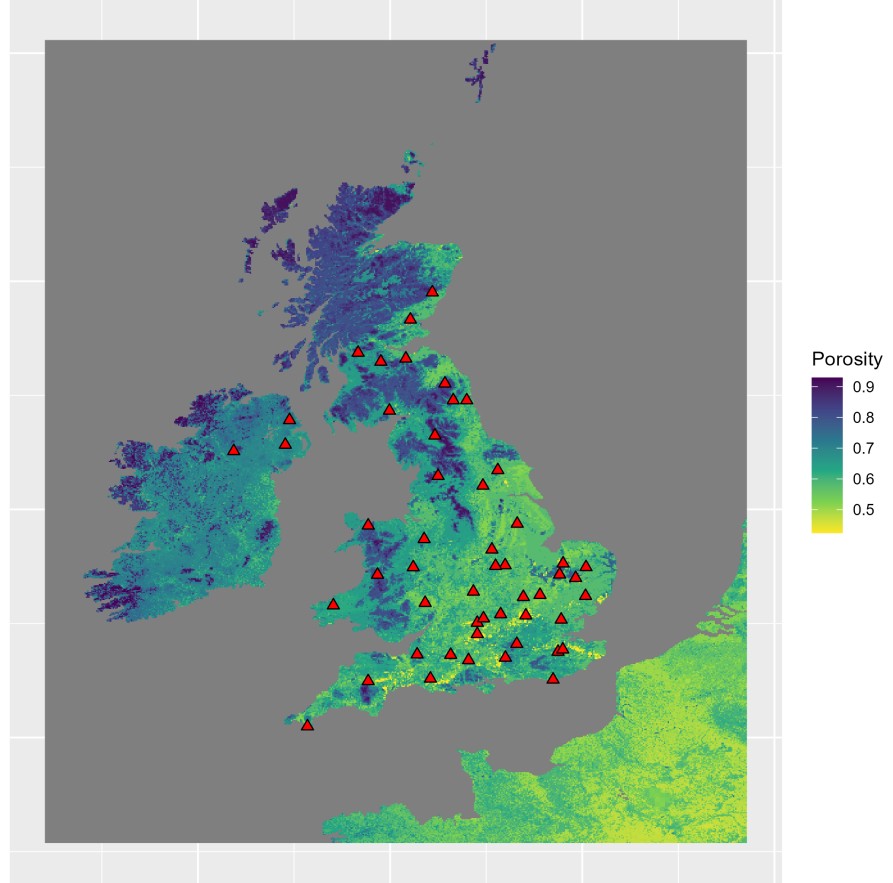

**Figure 1.** Map showing the COSMOS-UK network of sites measuring soil moisture (red triangles). The background colour scale shows soil porosity in $m^3\ m^{-3}$, equivalent to the volumetric soil moisture content at saturation.

### 2.1.3 Meteorological data

The main observations of precipitation are ultimately derived from the Met Office NIMROD system (Met Office, 2003),
which are assimilated into the Met Office UK Atmospheric High-Resolution Model for weather forecasting [NWP-UKV; Met Office (2016)]. The NIMROD system is based on a network of 15 C-band rainfall radars covering the UK (Harrison et al., 2000). This provides 2-km resolution composite data for precipitation rate every five minutes, from 2004 to the near-present. The assimilation system combines processed radar and satellite data, together with surface reports into the UK Met Office Numerical Weather Prediction (Milan et al., 2020). Bias corrections are well developed to improve absolute accuracy
of precipitation totals, although there may be further scope for improvement (Yu et al., 2020). The system provides the best combination of accuracy and spatial and temporal resolution available. Other meteorological variables for model development were also taken from the Met Office UK Atmospheric High-Resolution Model for weather forecasting.



### 2.1.4 SCAT-SAR

The Copernicus Global Land Services Scatterometer-Synthetic Aperture Radar Soil Water Index (SCAT-SAR SWI) is a fusion

of two satellite products (Bauer-Marschallinger et al., 2018): it uses high-resolution synthetic-aperture radar (SAR) surface soil moisture (SSM) data generated by the Sentinel-1 satellite mission, and combines it with the Advanced Scatterometer (ASCAT) SSM data from the MetOp satellites. The resulting product achieves both, high temporal and spatial resolution (daily, at 1 km over Europe), while providing improved reliability and accuracy, and is available from 2015 to the present day (Bauer-Marschallinger et al., 2018). In the SCATSAR-SWI algorithm, the ASCAT SSM and SAR SSM values are given a weighting

determined by the signal to noise ratio, specific to the time series at each grid point. Exponential moving averaging is then applied, similar to that described above for precipitation, except that a slightly different formulation is used, and a range of $\lambda$ values are used to vary the weight placed on current versus past data. We choose an intermediate value ($T = 15$ in their notation). The time series is normalised relative to the minimum and maximum SSM values recorded at each grid point; the scaling is therefore somewhat arbitrary and varies from location to location.

### 2.1.5 NRFA data

The UK National River Flow Archive (NRFA) contains daily data on river flow for 1212 gauging locations covering catchments across Great Britain (Fig. 2), along with corresponding catchment precipitation. Records span the period from the 1970s to the present day, although this varies among gauging sites. The gauging sites are classified into ~100 hydrometric areas, representing either single catchments or neighbouring hydrologically similar catchments. The data were obtained via the R package rnrfa

(Vitolo et al., 2016).

## 2.2 Model development

Our starting point was to build a simple statistical model which explains the temporal and spatial variability in the COSMOS observations of soil moisture. We treat $\theta'$ as the response variable, and use a hierarchical modelling approach (also referred to as "mixed-effects modelling") to account for the non-independence of data from the same sites (Bates et al., 2015). Beginning

with Eq. (3), we introduced variables to include the terms from the data sources listed above. The results of the model selection procedure are described below, but the parsimonious model identified is given by Eq. (4). This predicts volumetric soil moisture at time $t$ and site $s$, based on EMA precipitation, vapour pressure deficit $D$ and the SCAT-SAR Soil Wetness Index $I$ as:

$$\theta'_{ts} = \beta_0 + \beta_{\mathrm{P}} \langle P \rangle_{ts} + \beta_{\mathrm{D}} D_{ts} + \beta_{\mathrm{I}} I_{ts} + b_s + \epsilon_{ts}$$

$$b_s \sim N(0, \Psi)$$

$$\epsilon_{ts} \sim N(0, \sigma^2)$$

$$\theta_{ts} = \theta_{max,s} - \theta'_{ts} \tag{4}$$





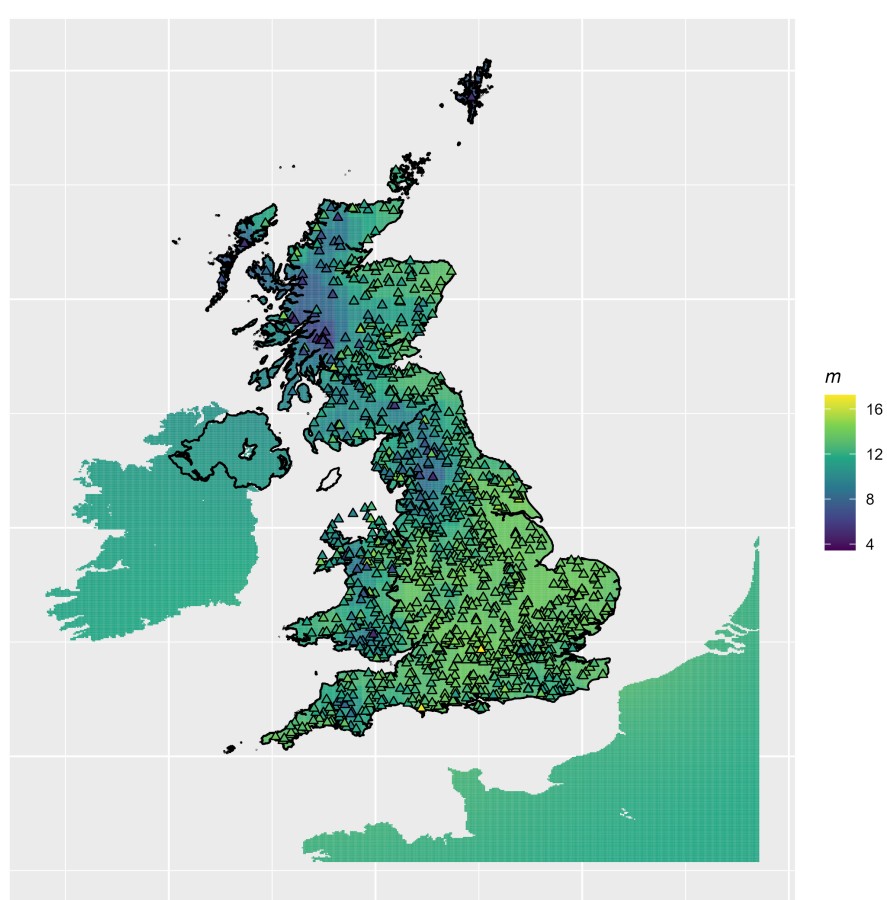

**Figure 2.** Map showing the National River Flow Archive sites measuring river discharge (triangles). The symbol colour shows the dimensionless slope $m$ of the relationship between the daily change in soil water storage $\Delta S$ and the daily change in EMA precipitation $\Delta \langle P \rangle$. The background colour shows the geostatistical extrapolation of $m$ from these sites to the wider domain produced by Bayesian kriging, shown on the same scale. The extrapolation to Ireland and continental Europe tends towards the mean because of the distance from the observation sites.





where $\beta_\mathrm{D}$ and $\beta_\mathrm{I}$ are the additional regression coefficients, and $b_s$ is the local deviation at site $s$ (a so-called "random inter-cept" term which accounts for the within-site correlation of residuals). The local deviations $b$ are assumed to be independently drawn from a normal distribution with mean zero and variance $\Psi$.

To improve the representation of spatial variation in $\beta_\mathrm{P}$, we used the NRFA data on river discharge and catchment precipita-tion. Whilst we cannot infer absolute soil moisture values in the catchment, we can estimate the *change* in soil moisture from
these data by calculating the quantity:

$$\widehat{\Delta S} = P - Q - E_\mathrm{pot} \tag{5}$$

closely correlated with $\Delta\theta = \Delta S/z$, but with evapotranspiration estimated at its potential rate $E_\mathrm{pot}$ rather than the actual rate. The $\widehat{\phantom{hat}}$ symbol denotes that this is an estimator of $\Delta S$, rather than the true value.

For each of the 1212 NRFA catchments, we calculated daily values of $\widehat{\Delta S}$, the exponential-moving-averaged precipitation
$\langle P \rangle$, and its day-to-day difference $\Delta\langle P \rangle$. The EMA filter linearises the relationship so that we can derive the slope representing how the change in soil water storage $\widehat{\Delta S}$ responds to change in (EMA) precipitation $\Delta\langle P \rangle$. For each of the 1212 NRFA catchments, we calculated this slope by linear regression i.e. fitting the linear model:

$$\widehat{\Delta S} = c + m\Delta\langle P \rangle. \tag{6}$$

The slope parameter $m$ is closely related to the $\beta_\mathrm{P}$ parameter in Eq. (4), as it relates the change in soil water storage to the
change in precipitation. Absolute values are not comparable as the effective depth $z$ is unknown, but the spatial patterns in both will be similar. We therefore use the spatial variation in $m$ from the 1212 NRFA catchments to estimate the relative spatial variation in $\beta_\mathrm{P}$ (Fig. 2). For each NRFA site $s$, we calculate the value:

$$\beta_{\mathrm{P},s} = \beta_\mathrm{P} \frac{m_s}{\bar{m}} \tag{7}$$

where $\beta_\mathrm{P}$ is the global (or "fixed effect") derived from Eq. (4), $m_s$ is the slope derived from Eq. (6) for site $s$, and $\bar{m}$ is the
mean value of $m$ across all 1212 sites.

## 2.3 Spatial extrapolation of $m$

The above equations provide values of $m$ for 1212 sites (Fig. 2), but we want to make predictions on a 2-km grid covering the UK. This is a common problem in the area of geostatistics, where we have limited measurement locations and we need to estimate values at many unobserved locations. The standard approach is known as kriging, a form of weighted local aver-
aging, where the estimates of values at unrecorded places are weighted averages of the observations. The kriging weights are calculated on the basis of the "semivariogram", which quantifies the form of the increasing variance (or decreasing covariance)





between pairs of points as the distance between them increases. Various mathematical models are used to describe the covariance as a function of this distance; the Matérn model is a common choice for its flexibility (Pardo-Iguzquiza and Chica-Olmo, 2008) in which the covariance is given by

$$C_\nu(d) = \sigma^2 \frac{2^{1-\nu}}{\Gamma(\nu)} \left(\frac{d}{\phi}\right)^\nu K_\nu \left(\frac{d}{\phi}\right) \tag{8}$$


where $\sigma^2$ is the variance, $\phi$ is a range or scale parameter, $\nu$ is a shape parameter, $\Gamma$ is the gamma function, and $K_\nu$ is the modified Bessel function. Graphically, this shows the scale at which values are highly correlated, and how this changes with spatial scale. Prediction at a new location is based on all the observations (within some cut-off distance), each weighted according to the degree of correlation at that distance predicted by the semivariogram. Kriging has been shown to be optimal

in the sense that it provides estimates with minimum variance and without bias (in the long-term statistical sense) (Cressie, 1990). Here, we applied the method with the Matérn covariance model, but in a Bayesian approach described below, using the geoR package for the R statistical software (Ribeiro Jr et al., 2020).

### 2.4 Variable selection, model fitting and application

For fitting Eq. (4), we used standard statistical model selection techniques (stepwise and best-subsets regression), which algo-

rithmically add and remove variables from the model, as well as manual variable selection. We assessed model performance using standard metrics: percentage variance explained, root mean-square error, and information criteria (Akaike Information Criterion or AIC, and Bayesian Information Criterion, or BIC)) to analyse which variables to include. Our aim was to select variables which makes sense in terms of the underlying processes, which maximise explanatory power, whilst considering their availability for near-real-time predictions at national scale and keeping the model as simple as possible. We used multiple $\lambda$

values to represent differential rates of water loss from different components within the pore size distribution. We tested models in which the contribution from different components with different $\lambda$ values was an explicit function of soil texture (i.e. interaction terms for $\langle P\rangle^{\lambda_1} \times \text{clayfraction} + \langle P\rangle^{\lambda_2} \times \text{siltfraction} + \langle P\rangle^{\lambda_3} \times \text{sandfraction}$). We also applied two machine learning techniques: random forests and extreme gradient boosting, both of which have been applied in similar contexts previously (e.g. Ramcharan et al., 2018) .

### 2.5 Bayesian approach

The parameters of mixed-effect and semivariogram models are typically estimated by maximum likelihood or ordinary leastsqaures methods. The parameters of mixed-effect models are typically estimated by maximum likelihood estimation. This estimates a single best estimate for the model parameters and their standard error. By contrast, the Bayesian approach explicitly attempts to quantify the probability distributions of the parameters, given the data and any prior information. This has the

advantages that it provides a robust means of estimating uncertainty on the parameters and predictions. This approach generally uses Markov chain Monte Carlo (MCMC) sampling, an iterative algorithm for calculating numerical approximations of multidimensional integrals. Many MCMC algorithms are available, and the mechanics of performing Bayesian statistical analysis



are described in several textbooks (e.g. Gelman et al., 2013). Here we use the Hamiltonian sampling algorithm, which provides a computationally efficient means of estimating the posterior distribution (Hoffman and Gelman, 2014; Betancourt, 2017;
Bürkner, 2018).

We also apply the Bayesian approach to the spatial extrapolation, so we recognise that the semivariogram is not a fixed, known quantity, but a geostatistical model with uncertain parameters. In brief, this means we account for the uncertainty in the variogram model, and represent each of the parameters as a probability distribution. Rather than assuming the variance is known, we calculate the posterior distribution of the parameters, given the observed data, and sample many realisations
of these to represent the uncertainty. We used the Bayesian kriging algorithm available in the geoR package (Ribeiro Jr. and Diggle, 2001)as described above, with uniform (uninformative) priors for the $\sigma, \phi$ and $\nu$ parameters. The algorithm makes some simplifying assumptions to speed computation time, such as discretising the prior parameter distributions.

The analysis formed a number of sequential steps.

1. We examined the variables needed to derive a model of temporal and spatial variation in $\theta$ at ~40 COSMOS sites. We
estimated the parameters of the final model by MCMC, starting with uninformative priors.

2. Having established the parsimonious model, we characterised the spatial variation in the $\beta_P$ parameter using the NRFA data on river discharge and catchment precipitation, and interpolated this on a 2-km grid across the UK using Bayesian kriging.

3. Finally, we applied the model to a 2-km grid across the UK to estimate the posterior distribution of predictions of $\theta$ each
day for 2015 to the present day. The model is run on a daily basis with the most recent data, typically ~7-10 days behind real time. We denote the data product "SMUK" for reference purposes.

4. We propagated the uncertainty that goes into the mapped predictions via Monte Carlo sampling. That is, we produced many simulations of each daily map, each one representing a sample drawn from the probability distribution for each parameter in the model and spatial interpolation process. The quantiles of the simulated values at each time and location
are used to create maps of credible intervals.

## 3   Results

### 3.1   Variable selection

Results of the variable selection procedure are shown in Table 1. The marginal $r^2$ describes the fraction of variation accounted for by the fixed-effect terms, and is calculated as the ratio of the variance of the fixed-effect components to the total variance,
where the latter is estimated as described by Nakagawa (2017).

The conditional $r^2$ adds in the variance accounted for by the $b_s$ terms, allowing variation in the intercept at each site, and is calculated as the ratio of the variance of the sum of fixed- and random-effect components to the total variance. The group-specific intercepts are useful in allowing more reliable estimates of the $\beta$ parameters, but are not used in prediction outside





the sample of COSMOS sites. RMSE is the root mean-square error, which estimates the average difference between the model

predictions and observations. AIC is the Akaike Information Criterion which measures model goodness-of-fit whilst penalising

complexity (number of parameters), where smaller values (more negative) indicate better performance. All the metrics show

the same pattern: model performance improves with the addition of terms for evaporation or vapour pressure deficit, as well

as the SCAT-SAR soil water index. Thereafter, adding terms for a range of variables representing soil properties, land cover,

and meteorgy shows no further improvement. Whether evaporation or vapour pressure deficit is used makes little difference,

as the two are closely related in the Penman-Monteith equation. Using vapour pressure deficit shows a slightly lower AIC

value and simplifies the model application, so this was preferred. Using multiple terms for EMA precipitation with a range of

values for $\lambda$ was also explored, and adding two additional covariates with higher and lower $\lambda$ values gave a slight improvement

in performance. Although this was deemed an improvement on the basis of AIC, we did not think the increase in model

complexity was merited in practical terms (less than 1 % difference in $r^2$ and RMSE). The asymptotic RMSE is 0.053 (i.e. 5.3

% in volumetric terms) so, on average, the model estimates are reasonably accurate.

The machine learning techniques (random forests and extreme gradient boosting) could produce very close fits to training data,

but their performance on independent test data was no better than Eq. (4), indicating that they were over-fitting.

## 3.2 Model checking

The five-year time series of the fitted model predictions for four sites is shown in Fig. 3. The model shows a reasonably good

correspondence with the COSMOS data. The model captures the main features of the observations: the seasonal cycle of wetter

winters and drier summers, the response to individual rain events, and the particularly dry summers of 2018 and (to a lesser

extent) 2019. The model inevitably has a smoothing effect, so does not capture all the extremes in the data.

Figure 4 shows the model predictions plotted against the observations for 37 of the COSMOS sites. The agreement is

generally good, with points falling around the 1:1 line. The figure shows that the model successfully captures the inter-site

differences in soil moisture. The amount of unexplained variation differs somewhat among sites, but there is no clear pattern

to this. In at least one case, the water level in the system of drainage channels is under direct control by the land managers, and

soil moisture can thus be driven strongly by a factor extrinsic to the model, rather than directly by precipitation. This site (and

three others with few records or very atypical data) were excluded from the analysis. The remaining 37 out of 41 sites shown

in Fig. 4 were used in the analysis, but the artificial manipulation of water levels may be a factor at other sites, such as the

Redmere (RDMER) site which is on an agricultural fen, where much of the observed change in soil moisture is not captured

by the model.

## 3.3 Spatial variation in $\Delta S / \Delta \langle P \rangle$

Figure 5 shows the daily change in soil water storage $\Delta S$ in response to the daily change in EMA precipitation $\Delta \langle P \rangle$ at 36

catchments with gauging stations in the NRFA data set, from nine different hydrometric areas. The relationships are strongly

linear over the whole range in $\Delta \langle P \rangle$, though with more scatter at some sites than others. Data for 1212 catchments are available,

and show similar patterns to the examples plotted here. There is variability between catchments in the slope of this relationship,



**Figure 3.** Time series of volumetric soil moisture $\theta$ at four of the COSMOS network sites. The daily mean of the COSMOS observations are shown, together with model predictions from Eq. (4). Sites cover a range of soil types from across the UK - CRICH: Crichton, Dumfries; LIZRD: The Lizard, Cornwall; NWYKE: NorthWyke, Devon; PLYNL: Plynlimon, Ceredigion.





**Figure 4.** Volumetric soil moisture at 37 of the COSMOS network sites predicted by the Eq. (4), $\theta_{\mathrm{pred}}$, plotted against soil moisture observed by the cosmic-ray neutron sensors, $\theta_{\mathrm{obs}}$. The 1:1 lines are shown. Site codes are given in Stanley et al. (2019).

and this provides empirical landscape-scale information on the water retention of the soils in these catchments. Wetting of soil during rainfall is almost instantaneous, so the slope of this relationship is dictated by the loss of soil water after rain events by drainage and evaporation. Low values of this slope $m$ occur indicate that the soils are slow to drain and dry.

This variability is shown in Fig. 2, and shows the lowest values of $m$ occur in the NW of Scotland, with higher values in the south and east of England. There is a strong correspondence between this map and the map of porosity (Fig. 1), even though these are entirely independent data sources. The causal nature underlying this correspondence is potentially complex, but soils with high porosity have a greater capacity to store water (by definition), so may be less prone to short-term run-off. Porosity is strongly correlated with organic matter content, and organic matter is known to increase soil water holding capacity

and water retention (Lal, 2020). However, the relationship may not be straightforward (Minasny and McBratney, 2018; Weiss





**Figure 5.** Daily change in soil water storage $\Delta S$ in response to the daily change in EMA precipitation $\Delta \langle P \rangle$ at 36 catchments with gauging stations in the NRFA data set. Each catchment is shown by a different colour. Catchments are grouped into panels for nine different hydrometric areas, representing sites from either the same watershed or neighbouring hydrologically similar catchments. Because many point overlie each other, symbols are partially transparent to aid visbility. Best-fit lines for each catchment are also shown.





et al., 1998). Because of the dense coverage of NRFA gauging sites, kriging gives a very reasonable interpolation of the data in Fig. 2. NRFA data only cover Great Britain, so extrapolated values in Ireland and continental Europe tend towards the mean.

## 3.4 Model application

The interpolated values of $m$ were used to estimate values of $\beta_{\mathrm{P},s}$ according to Eq. (7), for each 2-km grid cell in the NWP-
UKV model domain. These values were used in Eq. (4), along with the soil porosity data, Met Office precipitation and vapour pressure deficit data, and the SCAT-SAR soil water index described above, to predict daily soil moisture for the period 2016 to present day (2023). The spatial pattern in mean soil moisture clearly reflects the patterns in the inputs and soil parameters (Fig. 6), which are themselves strongly correlated. For example soil porosity and precipitation show a strong correspondence, because of the relationship between precipitation and peat formation, and the effect of organic matter on bulk density and hence
porosity. The exception to this is the spatial pattern in the satellite SCAT-SAR SWI, which, because of the rescaling, shows only where the mean sits in relation to the maximum and minimum values at each location, usually close to 0.5 if variation is symmetrical. The temporal variation is still useful, but the spatial variation in the absolute values is not informative.

Figure 7 shows the predicted response of soil moisture to bands of precipitation from weather fronts on 8th and 9th March 2023. Precipitation fell mainly in the south-west on 8th March, becoming more widespread over the southern half of England
and Normandy on 9th March. The resulting spatial pattern in elevated soil moisture is clear on the subsequent days. Northern Scotland had heavy precipitation in the preceding days, which remained wetter than average over the course of the four days shown. Despite being a simple one-line equation, the model captures the highly dynamic and spatially-detailed response of soil moisture to precipitation. The spatial and temporal dynamics are better visualised in a movie format, and animated visualisations of predictions for each year are available via the data repository (https://gws-access.jasmin.ac.uk/public/dare_uk/smuk/anim/).

## 335 3.5 Discussion

The results show that a simple linear model can capture the temporal dynamics in soil moisture observed at the COSMOS sites. In terms of temporal variation at a given site, this model could reproduce the same behaviour as a more complex model with explicit representation of soil hydraulic processes: the agreement with the observations (0.05 m$^3$ / m$^3$ RMSE) is better than that reported for the process-based JULES (0.18 m$^3$ / m$^3$ RMSE, Pinnington et al., 2020), and considerably better than the
satellite-based products reported by Peng et al. (2021) (noting the different definition of RMSE used there). The EMA filtering of the precipitation time series is a simple but key step in achieving this simplification. We anticipated needing multiple $\lambda$ values to represent different components within the pore size distribution, but the effect of this was minimal compared to using a single $\lambda$ value. Although models using multiple $\lambda$ values could explain more variation at a given site, this did not produce a model with better predictive power across sites. Simlarily, models in which the contribution from different components with different $\lambda$ values was an explicit function of soil texture did not show any improvement. It appears that the effects are not easily
generalisable across different soils, probably because the hydraulic properties depend on more complex phenomenon (such as pore size distribution, layering, macropore connectivity, hydrophyllic/hydrophobic properties of organic matter etc.), so are not easily predictable purely from soil texture (Walczak et al., 2002; Wang et al., 2022). There may be similar phenomena in the



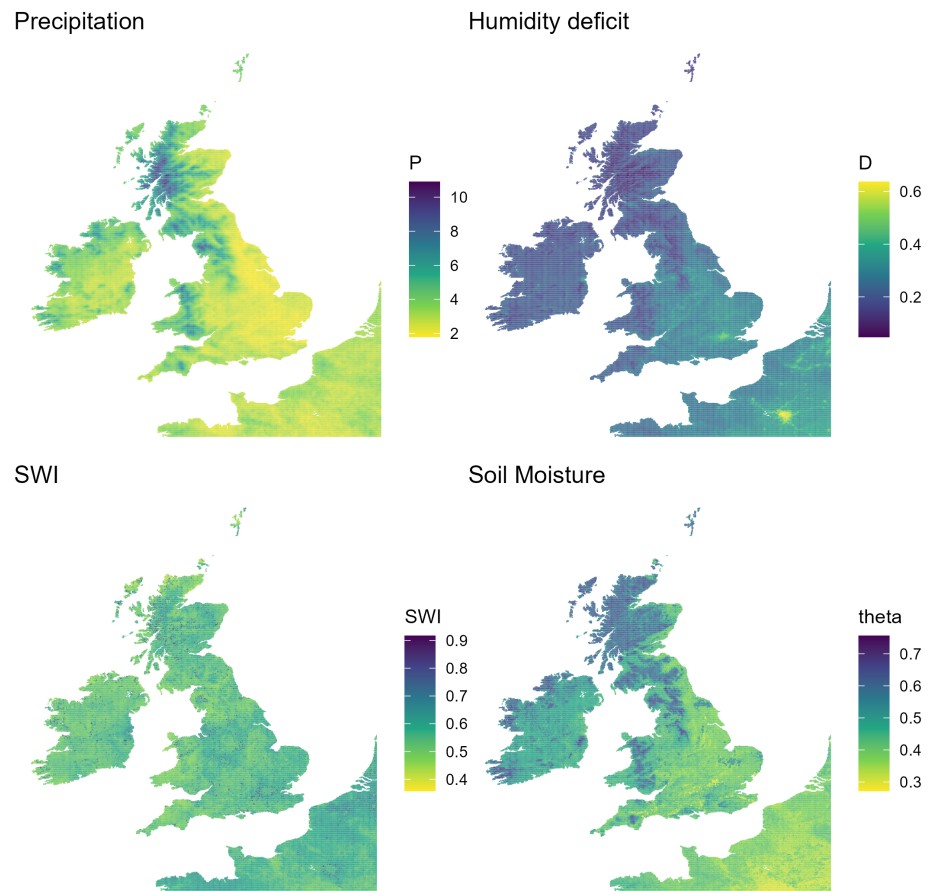

**Figure 6.** Maps of model inputs (precipitation $P$ in mm/d, vapour pressure deficit $D$ in kPa, and SCAT-SAR soil water index (SWI, dimensionless)) and predicted volumetric soil moisture (m$^3$ / m$^3$) averaged over 2016-2022.

time domain, for example if extreme drying causes the soil structure to change, so that the response to rainfall is no longer

linear, or changes over time. From a modelling perspective, these phenomena effectively a constitute stochastic error term, because they are not feasibly predictable from the information that is available at the scale of interest.

     Most of the co-variates explored for inclusion in the model are strongly correlated, so a clear set of variables to use is not obvious. Our choice was an attempt to be parsimonious, using the fewest variables where the causal relationship is clear. Slightly better fits could be obtained using more variables, but this runs the risk of over-fitting (Gelman et al., 2020) and

complicates application of the model, as accurate estimates of all variables need to be available over the domain for predictions. The machine learning techniques (random forests and extreme gradient boosting) could also produce closer fits to training data, but tended to over-fit, and their performance on independent test data was no better. Therefore the standard linear mixed-effect models are preferred because of their interpretability and ease of uncertainty propagation.





**Figure 7.** Predicted course of soil moisture over four days from 8th to 11th March 2023, in response to precipitation from weather fronts passing over from the south-west. Soil moisture is shown as the difference from the annual mean soil moisture for the year at each location ($\Delta SM$), with blue indicating wetter than the mean. The total precipitation on each day is shown by the green contours.



Although we have very good data on soil moisture at ~40 sites across the UK, this in itself does not strongly constrain the
national-scale map of soil moisture at a given time; the area sampled is tiny in relation to the area of the UK, and soils vary
widely over this domain. Since we know the inputs to the system rather accurately (from the rainfall radar data), as well as
the meteorological variables governing evaporation, the big uncertainty is in the spatial variation of the parameters that govern
soil water retention. Existing data sources (e.g. HOST (Boorman et al., 1995), HWSD (FAO, 2012), SoilsGrid (Hengl et al.,
2017)) provide rather uncertain estimates because they are themselves extrapolations from a relatively small number of soil
cores, and are often at variance with each other because the true spatial distribution of these parameters is unknown. Here, we
address this problem in two ways. Firstly, we use remotely-sensed data from SCAT-SAR, which gives direct information over
the whole domain on an appropriate time scale, and integrates effects of both gains and losses of water, as the signal responds
directly to surface soil moisture content. However, the data are only on a relative scale specific to each location, so are not
easily interpreted as absolute changes in soil water or as a spatial pattern. Secondly, we use a novel approach, using water
balance data to make inferences about the soil water retention parameter $\beta_P$. This brings in a vast amount of data (more than
1000 catchments, over several decades at daily resolution) where we have direct measurements of inputs and outputs of water
from the catchments. Differences in the response across catchments allows us to infer spatial differences in the hydrological
properties far more directly than, for example, using global soils databases and modelling hydraulic properties as a function
of estimated soil texture. Basic analysis of the dynamics of a first-order "bucket" hydrological model shows that $\Delta S / \Delta \langle P \rangle$
will be linearly related to $\beta_P$, so provides a valid estimator of the spatial variation in the latter. The approach thus integrates
the net effect of the hydraulic properties of the soils within a catchment that we want to capture in $\beta_P$. The finding that the
spatial pattern in the soil water retention parameter derived in this way strongly reflected the pattern in soil porosity suggests
the approach is discerning real hydrological patterns in the landscape, given the complete independence of these data sets. This
gives us further confidence that the approach is valid. Further data sets could be assimilated into the method in future, including
historical conventional soil neutron probe measurements (Bell et al., 2022) and on-going monitoring of lake levels.

An important advantage of the model developed here is its simplicity, and consequent speed and ease of computation.
Process-based hydrological and surface energy balance models typically have thousands of lines of code, and to simulate a
large domain over a number of years may take days of computation time. Our model is effectively a single equation, and so
computation time is insignificant. Unlike machine learning approaches, the linear model retains interpretability because the
parameters have a physical meaning, and suffers less from potential over-fitting.

A further advantage over machine learning approaches is that the Bayesian approach allows uncertainty to be propagated in
a coherent way (van Oijen, 2020, 2017), although work to combine these approaches is ongoing (e.g. Kirkwood et al., 2022).
Because we have estimated the posterior distribution of all the model parameters, including the parameters of the geostatistical
model which interpolates $\beta_P$, we can draw samples from these to produce the posterior distribution of predictions. That is, for
each location and timepoint, we can estimate soil moisture (say) 1000 times using sets of parameters from across the posterior
distribution, and the spread provides the correct uncertainty associated with our predictions. Although it is comprehensive in
dealing with uncertainty, Bayesian approaches can be demanding in terms of computation time, and this typically limits their
application to simpler models such as the one developed here, rather than complex process-based models. Even here, the data





volume involved in 1000 realisations of the predicted soil moisture maps is still considerable. The computation time for the
Bayesian kriging of the $m$ parameter is around 24 hours for the full data set (running R version 3.6.3 on Linux CentOS7),
but this is essentially a one-off operation, and only needs repeating when there are substantial updates to the NRFA data. The
computation time scales with $n^2$, and good approximations are obtained using subsets of the data, with computation time of
less than one hour. Currently, we estimate the model parameters with the data over a fixed past-time period for both COSMOS
and NRFA. In principle, we could update the model parameters on a daily basis, as new data become available. However, the
marginal effect of adding a single day of data is not large enough to be worthwhile, and annual updates are more reasonable.

    As with any modelling study, we have to make some pragmatic choices and simplifying assumptions. Perhaps the most
significant of these here is to take the soil porosity map as a known quantity. While we account for the fact that the map of $\beta_\mathrm{P}$ is
uncertain, because there is no reliable pre-existing data source, we effectively assume that the soil porosity map is known with
complete certainty. In reality, this is clearly not the case, and could be addressed in future work. Similarly, because the NRFA
data only cover Great Britain, estimates of $\beta_\mathrm{P}$ in Ireland and France are poorly estimated by extrapolation from the GB data
points, so tend towards the mean. A better approach would be to use some other covariates in the geostatistical modelling, if
predictions outside the region with observations are required. Along these lines, COSMOS observations (Bogena et al., 2022)
and all the inputs variables are available for continental Europe except NRFA, so the approach could be extended to the wider
domain in future, if this issue is addressed. Whether the same simplifying assumptions would be reasonable over a wider region
with drier climates would require investigation.

    Model outputs can be expressed as volumetric soil moisture or as water-filled pore space (WFPS). The latter is seen as more
relevant to biological process as it is a more obvious proxy for soil oxygen content. Model outputs in this form are currently
being used to model emissions of greenhouse gases and radon from soils in the UK. For water uptake by plants, the soil water
potential is the key variable, and estimating this from volumetric soil moisture requires all the additional information needed
to characterise the soil water retention curve, and this remains a challenging task.

    One area where the model will fail is where water levels are managed, either by irrigation or controlling lateral flow of water
in ditches. The model currently assumes that all inputs of water come only from precipitation, and results will not be reliable
where this is not the case. This does not apply to large areas in the UK, but the question of irrigation demand is an important
application of soil moisture modelling generally, and not currently covered here. Some representation of the additional water
inputs would be needed to address this.

## 4   Conclusions

This study used a statistically-based approach to estimate soil moisture for the UK. The focus was on accurately predicting
absolute values of soil moisture, closely tied to direct measurements, to address the perceived weakness of remotely-sensed data
products. We investigated the influence of different ground-based, modelled, and remotely-sensed variables on the accuracy
of soil moisture estimates by analyzing observations from a network of 40 sites with cosmic-ray neutron sensors. The spatial
variation in the parameter describing the soil water retention (and thereby the response to precipitation) was estimated using





daily water balance data from ~1200 catchments with good coverage across the country. We used a Bayesian approach to allow the uncertainty from the calibration at 40 sites to be propagated through the extrapolation on a grid covering the whole UK.

The main findings of this work are as follows:

We found that the parsimonious model was a linear model using the exponentially-weighted moving-average precipitation, the remotely-sensed SCAT-SAR index, the vapour pressure deficit (as a close proxy for evapotranspiration) and soil porosity. The simple linear model could emulate the behaviour of a more complex process-based model, and performed better in reproducing the COSMOS observations.

The spatial pattern in the soil water retention parameter derived from catchment water balance strongly reflected the pattern

in soil porosity; this gives us confidence that the approach is discerning a true pattern in the hydrological properties of the soils, rather than an artefact in the data.

Given the high resolution of the inputs in time and space, the model can predict the very detailed variation in soil moisture which arises from the sporadic nature of precipitation events, including the small-scale and short-term variations associated with orographic and convective rainfall. Predictions over the period 2016 to 2023 demonstrated realistic patterns following the

passage of weather fronts and prolonged droughts.

The model has negligible computation time, and inputs and predictions are updated daily, lagging approximately one week behind real time.

*Code and data availability.* Model output from 2016 onwards will be available from https://eidc.ac.uk/, URL to be confirmed. Model output is currently available in GeoTIFF and AVI movie formats via https://gws-access.jasmin.ac.uk/public/dare_uk/smuk/.

*Team list.* Joshua Alton, Anne Askquith-Ellis, Emma Bennett, James Blake, Milo Brooks, Nicholas Cowan, Hollie Cooper, Jonathan Evans, Matthew Fry, Duncan Harvey, Helen Houghton-Carr, Alan Jenkins, Sarah Leeson, William Lord, Gemma Nash, Daniel Rylett, Peter Scarlett, Simon Stanley, Oliver Swain, Magdalena Szczykulska, Simon Teagle, Emily Trill and Alan Warwick. The complete past member list of the COSMOS-UK team can be found at https://cosmos.ceh.ac.uk/about-team.

*Author contributions.* PL performed the modelling analysis and wrote the manuscript; the COSMOS-UK team provided the COSMOS data

on which the model is based.

*Competing interests.* The authors declare no competing interests.



**Table 1.** Model selection criteria for linear mixed-effect models with a range of variables included. Marginal r2 represents the variance explained by the fixed effects; conditional r2 is interpreted as a variance explained by the entire model, including both fixed and random effects. Both are calculated according to Johnson (2014). RMSE is the root mean-square error, which estimates the average difference between the model and the observations. AIC is the Akaike Information Criteria. Abbreviations for variables examined were P: Precipitation over previous 30 days with exponential moving average applied; E: potential evapotranspiration rate calclulated using the Penman-Monteith equation; D: water vapour pressure deficit; SWI: soil wetness index from the SCAT-SAR satellite product; TA: air temperature; Pi + ... + Pn: EMA precipitation with a range of $\lambda$ decay rates applied; Land cover type is the site classification according to the UKCEH Land Cover Map 2019; HOST: Hydrology of Soil Types classification from Boorman (1995); TWI: topographic wetness index (Beven and Kirkby, 1979). Clay, sand, silt and coarse fragments represent the soil texture components from Ballabio et al. (2016). AWC: available water-holding capacity estimated from the above soil texture components from Ballabio et al. (2016).

| Model variables | Marginal r2 | Conditional r2 | RMSE | AIC |
|---|---|---|---|---|
| EMA rainfall (P) | 0.359 | 0.469 | 0.072 | -136280.6 |
| P + E | 0.480 | 0.593 | 0.063 | -147852.1 |
| P + D | 0.478 | 0.590 | 0.063 | -150693.8 |
| P + E + SWI | 0.598 | 0.692 | 0.053 | -166749.7 |
| P + D + SWI | 0.598 | 0.688 | 0.053 | -169882.0 |
| Pi + ... + Pn + D + SWI | 0.605 | 0.697 | 0.052 | -171176.5 |
| P + D + SWI + TA | 0.598 | 0.688 | 0.053 | -169867.1 |
| P + D + SWI + Windspeed | 0.599 | 0.693 | 0.053 | -169344.6 |
| P + D + SWI + Soil carbon content (C) | 0.598 | 0.689 | 0.053 | -169877.6 |
| P + D + SWI + HOST class | 0.597 | 0.693 | 0.053 | -169781.7 |
| P + D + SWI + Bulk density | 0.597 | 0.689 | 0.053 | -169875.7 |
| P + D + SWI + TWI | 0.608 | 0.692 | 0.052 | -168037.3 |
| P + D + SWI + Land cover type | 0.643 | 0.695 | 0.053 | -169788.5 |
| P + D + SWI + Latitude * Longitude | 0.597 | 0.690 | 0.053 | -169847.9 |
| P + D + SWI + Soil type | 0.593 | 0.692 | 0.053 | -169850.2 |
| P + D + SWI + AWC | 0.597 | 0.688 | 0.053 | -169880.6 |
| P + D + SWI + Coarse_frag | 0.614 | 0.689 | 0.053 | -169875.8 |
| P + D + SWI + Clay | 0.595 | 0.686 | 0.053 | -169868.7 |
| P + D + SWI + Sand | 0.596 | 0.688 | 0.053 | -169865.8 |
| P + D + SWI + Silt | 0.597 | 0.689 | 0.053 | -169866.4 |



*Acknowledgements.* This work was supported by the Natural Environment Research Council award number NE/R016429/1 as part of the UK-SCAPE programme delivering National Capability, and award number NE/S003614/2 "Detection and Attribution of Regional greenhouse gas Emissions in the UK (DARE-UK)". We acknowledge the provision of the UK Meteorological Office (UKMO) data for the purposes of the UK-SCAPE programme.



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
