# Peer review of "Mapping soil moisture across the UK: assimilating cosmic-ray neutron sensors, remotely-sensed indices, rainfall radar and catchment water balance data in a Bayesian hierarchical model"

_EGUsphere, 2023_

## Author Response (AR1)

**Reply to referees' comments**

Peter Levy

2024-07-26

We thank the referees for their time taken. Their comments are shown in *italics*; our response is beneath in normal font.

**Referee 1**

*The authors have done a job on high-resolution soil moisture modeling at the UK scale. The paper is well structured, but a major revision is needed before publication. My main issues include:*

*1. To add a flowchart that systematically shows the various parts of the study and the roles of the various data.*

- A good idea - we added this as a new Figure 1 in the revision.

*2. To add a description of the matching of COSMOS sites to model grids. It is not clear at this point how to match COSMOS data at nearly 100m resolution with models at 2km resolution.*

- This is straightforward because the COSMOS sites were simply matched to data for the grid square in which they were located. We have stated this in the revision.

*3. As the authors said, they used decades of stream flow data. Have these watersheds changed over the last few decades? In particular, are there any hydraulic structures or water extraction projects conducted during this period? How would these decades of river flow data affect the results of this study if they are unsteady?*

- Where these do occur, it would indeed make a step change in the parameters we are estimating. The NRFA data include meta-data on any known man-made changes of this kind, and we have tried to remove data prior to these changes where they have occurred. However, of the 1200+ catchments, this affects relatively few, most of which have been identified and removed, so we do not think this is a major problem with the analysis. We have added text to this effect in the revision to the manuscript.

*4. The information presented in Fig.3 is not clear, please revise it. Please add the corresponding rainfall. Please show the soil moisture of one or two months in different seasons.*

- We have added rainfall to the figure 3, as well as the satellite index for comparison. Showing some contrasting months as separate plots loses the broader picture and would require an extra figure. We think a better compromise is showing a single year, where we see some detail at the level of individual rain events, but retain a picture of the seasonal pattern.

**Referee 2**

*1. As a key motivation for inventing a completely new hydrological model, I am missing an extensive introduction of existing hydrological models, their methods and capabilities to predict spatial SM in the UK, where and why they fail, and what will be done differently in this study to solve these issues. Has nobody before operated a hydrological model in the UK? What is their resolution? Has nobody before integrated discharge data? Or satellite data? Or CRNS data? There is plenty of literature here that needs to be discussed before it becomes clear whether you actually invented a completely new approach or took or ammended parts of existing ones. And whether this choice is adequate compared to the performance of existing models.*

- We accept this point, and have added text to the introduction on existing soil moisture products.

*2. The authors present their "simple" model with a number of unclear assumptions (Lines 58-70). E.g., treating soil moisture dynamics as a pulse-decay curve with exponential shape. I have strong doubts that this is a valid assumption for soil hydrological processes, neglecting porosity, capillary forces, van.Genuchten models, vegetation influence, etc. If the authors are really convinced about their assumptions here, the reader would at least expect scientific argumentation of why these assumptions hold, e.g., using insights from existing literature. The whole section hardly names any hydrological paper to strengthen the choice of assumptions, which would be OK for the first hydro model invented in 1950, but not in 2023.*

- We find this a strange comment. The assumptions are explicit in these lines and in the equations, as well as in the referee's comment itself. We are not "neglecting porosity, capillary forces …" but demonstrating that they do not need to be represented explicitly: at a given site, the dynamics can be summarised very simply as exponential decay, and thereby linearised via the EMA filter. We cite three hydrological papers which have used the same approach successfully. We could add a section which demonstrates how the EMA linearises the soil moisture response to rainfall, but this is basic mathematics and does not require citation of hydrological papers - it follows from first principles. We could add this in supplementary information perhaps if the editor thinks this would be helpful.

*3. A major challenge when comparing soil moisture from hydrological models and COSMOS data is the vertical soil moisture profile. COSMOS averages soil moisture between 0 and 80 cm, with an exponential weight which is higher for shallower layers and that depends (unfortunatelly) on the soil moisture profile itself. It changes over time. And it is not trivial to what layer of the hydrological model these measurements should be compared to, and how. Many other papers have addressed this challenge already. While in the present paper, I cannot find any hint on how exactly the authors compared observed and predicated soil moisture layer-wise. Please elaborate.*

- We now explain the depth sensitivity of COSMOS measurements, and we make the point explicitly in the revision - that the observations (and thus predictions) are subject to this varying-depth effect, and there is no simple solution to this. With additional profile measurements, one could estimate the depth profile sensitivity (Scheiffele et al. 2020) so as to normalise estimates of soil moisture to a constant depth, but the necessary observations are not currently available. At no point do we say that there are any "layers of the hydrological model", and the equations are explicit.

*4. The agreement between observed and predicted soil moisture does not look convincing to me (Fig. 3 and 4). There are obvious biases and unmatched dynamics still visible. Performance metrics like KGE or R² are missing to assess the qualitiy of the prediction. The RMSE alone could miss important differences in dynamics.*

- r2 for every model variant is listed in Table 1, along with AIC as the more useful measure of comparative goodness-of-fit. The conditional r2 is 0.69 for the model we selected to use operationally; sure, the agreement is not perfect, but the point is that the simple linear model does better than the previous satellite estimates and the more complex models cited. Whatever biases and unmatched dynamics that remain are not explicable by any of the variables that are available.

*5. I wonder whether the performance of the model has been tested on uncalibrated sites. A usual approach to test spatial extrapolation or regionalization models is to train them on a few sites and test them on other sites. Please add such an analysis such that the reader can assess the reliability of your high-resolution model at sites other than the COSMOS sites.*

- We are not averse to adding cross-validation in principle, but it doesn't achieve anything additional. The point of the hierarchical approach is that it treats the site-to-site variability explicitly, and estimates the global parameters having accounted for this. So in principle, we can already say how well we expect the model to do at a new site, since we have estimated the variance $\phi$.
- One real advantage of this approach is that we can propagate this uncertainty that we know will arise at each new site into the predictions. Cross-validation is a more computationally intensive way to quantify that same site-to-site uncertainty, but does not provide an easy means of propagating that uncertainty into predictions. The strength

of AIC is that, in theory, it provides a measure of out-of-sample prediction, so indicates which model should give the best prediction at sites outwith the calibration set.

*6. The major selling point of the new model seems to be computational speed (Line 381). However, there are other hydrological models which are also based on simple principles, physical parameters, and still extremely fast. One of many examples could be the mHM model (Samaniego et al. 2010), proofed to be one of the best hydro models globally. A major difference is that they regionalize the calculation of soil porosity, while your model takes a given map for granted. It would be important to highlight the differences to this and other existing models in terms of methodology, speed, and quality of results.*

- We have added some discussion of other modelling approaches which have been applied in the UK to the introduction. One obvious difference with the MHM is the degree of complexity, since it is a system of ODEs with at least 62 parameters to be estimated, rather than a single linear equation with six parameters (Eqn 4). As an aside, the MHM paper referred to appears to do something similar to the method we describe here, albeit using very different terminology (e.g. "regionalisation").

*# Minor concerns 1. The structure of the introduction is unconventional and confusing. It appears that the introduction has not ended before section 1.1, but the subsequent description of the hydro model used seems also be part of the introduction, too. After that, the aims of the study are outlined two pages later. This is highly confusing and should be changed. Section 1.1., and maybe parts of 1.2, should move to the methods section. Please elaborate on the structure and outline of the study at the end of the introduction. I was not able to identify a clear hypothesis, other than making "the most accurate estimate of mapped soil moisture as possible", which is both vague and nonscientific language.*

- By contrast, referee 1 says "the paper is well structured". We explain the problem, then introduce our approach to modelling soil moisture in time (1.1) and in space (1.2), and give explicit aims (1.3). The aims only make sense in terms of the problem we are trying to solve (making accurate maps of soil moisture) and our approach to solving it (integrating disparate data sources in a linear model), so inevitably appear later. We are not testing any hypothesis here because we are not doing an experiment. There is nothing "vague and nonscientific" about our stated aims.

*2. The introduction seems to be a bit biased, as no issues of the CRNS technique have been addressed, while many issues of remote sensing products are prominently mentioned. Especially since the argumentation focuses towards the unwanted influence of vegetation water and soil properties, it is necessary to indicate that CRNS has very similar issues, as it does not work reliably in highly vegetated, highly prorous, or highly organic soils (Bogena et al. 2013, Rasche et al. 2021, etc).*

- This is true, but CRNS sites are usually selected to avoid these problems, and particular issues (e.g. vegetation mass) can be accounted for by ancilary local site measurements;

this is not feasible with remote sensing. We have pointed out the weaknesses in CNRS and cited the suggested references, but we do this in the Discussion where it fits better.

*3. Section 2.1.1.: A proper and unbiased introduction of the COSMOS technique, which is, as was advertised, key to this study, requires more description of the pros and cons. In that sense, the description is actually incomplete. Neutrons are not only sensitive to soil moisture, but to any hydrogen pool in organic matter, vegetation, snow, etc. This is a highly relevant information to assess the performance and quality of your results. Also the fact that COSMOS data is calibrated on actual soil moisture is very relevant, because neutrons are a relative quantity just as the remote sensing data you critisize. Furthermore, Köhli et al. speaks of 15 to 80 cm of sensing depth, why do you mention max. 30 cm depth here? The answer is the wet soil in UK, which brings us back to the fact that limitations of COSMOS have not been properly explained here. Please elaborate on the quality of the CRNS data and provide related citations.*

- Same point as #2 above. We will some text to give better balance as the referee suggests.

*# Specific comments*

*## Abstract:*

*1. The abstract is not logical or at least unclear. You motivate your study by the fact that remote-sensing data, soil hydrological data and vegetation introduce uncertainty. Then you present a solution which involves a remote-sensing product and soil properties. The reader would expect a brief argumentation why this solution solves the previously mentioned issues while it again makes use of them.*

- The point we failed to make was that our method reduces uncertainty by integrating multiple data sources, all of which have weaknesses, but together act as a better constraint on the true soil moisture. We have edited the text to this effect.

*2. The study was further motivated with the fact that remote sensing data have issues to provide absolute soil moisture. The solution presented, however, seems to be good at explaining variation only, with no mention of absolute SM predictions anymore (at least in the abstract). If you raise an issue in the beginning, the reader would expect a reference to it at the end of the story.*

- We have edited the abstract so that it is explicit that the absolute agreement with observations is better than comparable simulations with process-based models.

*3. Please use scientific and more concrete language when describing the models used. A "simple model", as the major outcome of your study, is not an adequate description. Can you name it? Is it a statistical or bucket model? Help the reader to categorize the key model of your study among the many existing model variants in hydrology. Similarly, please name or briefly elaborate on "a process-based model" which you mentioned using as a benchmark.*

- We have replaced this with "linear statistical model", since it is widely understood what this means.

*4. The last sentence does not make sense to me. If there is neglible computation time and assimilation of realtime data, why it lacks behind one week?*

- The referee has misread the sentence. We do not say "assimilation of realtime data". We say "predictions are updated daily, lagging approximately one week behind real time"; it takes about a week for the weather and satellite data to become available. Computation time is <5 seconds for the whole domain, once the input data are available.

*## Manuscript*

*Line 26: Consider mentioning also the useful integration depth of this measurement technique.*

- We have added text to this effect.

*Line 33: Can you assign the individual citations to each problem separately, instead of lumping them all at the end of the sentence? Thanks!*

- There is not a one-to-one correspondence between the problems and the citations; most address several of these issues.

*Line 36: replace "are" by "and" (…influenced)*

- No, the "and" is on the next line. "are" is correct here.

*Line 295: "there is no clear pattern to it". Please rephrase. The interpretation of the pattern is scientific research. Just because no reason for the variations has been identified so far, it does not mean that there is no reason or no underlying pattern at all.*

- We do not say there is "no reason for the variation", we merely say "there is no clear pattern to it" so we cannot interpret it information available to us.

*Code availability: it is highly recommended to publish the model code, e.g. in a git repository, as it is common standard for other hydrological models.*

- We could publish the code as suggested, but since it is a linear statistical model, it is only a single line of R code, so this seems superfluous. Most of the code we wrote is data wrangling to change between formats and data structures for the inputs, so very task-specific and not very interesting, but happy to make public on GitHub. Unfortunately the meteorological data used is not open-access, so we can't provide a live working version, though we provide the outputs in this way.

References

Scheiffele, Lena M., Gabriele Baroni, Trenton E. Franz, Jannis Jakobi, and Sascha E. Oswald. 2020. "A Profile Shape Correction to Reduce the Vertical Sensitivity of Cosmic-Ray Neutron Sensing of Soil Moisture." *Vadose Zone Journal* 19 (1): e20083. https://doi.org/10.1002/vzj2.20083.